# Take Me Home, Protein Roads: Structural Insights into Signal Peptide Interactions during ER Translocation

**DOI:** 10.3390/ijms222111871

**Published:** 2021-11-01

**Authors:** A. Manuel Liaci, Friedrich Förster

**Affiliations:** Bijvoet Centre for Biomolecular Research, Structural Biochemistry Group, Utrecht University, Universiteitsweg 99, 3584 CG Utrecht, The Netherlands; a.m.liaci@uu.nl

**Keywords:** signal peptide, signal peptidase, ER translocon, endoplasmic reticulum, protein targeting, chaperones, protein translocation

## Abstract

Cleavable endoplasmic reticulum (ER) signal peptides (SPs) and other non-cleavable signal sequences target roughly a quarter of the human proteome to the ER. These short peptides, mostly located at the N-termini of proteins, are highly diverse. For most proteins targeted to the ER, it is the interactions between the signal sequences and the various ER targeting and translocation machineries such as the signal recognition particle (SRP), the protein-conducting channel Sec61, and the signal peptidase complex (SPC) that determine the proteins’ target location and provide translocation fidelity. In this review, we follow the signal peptide into the ER and discuss the recent insights that structural biology has provided on the governing principles of those interactions.

## 1. Introduction

The secretory pathway is a protein trafficking highway utilized by more than a quarter of the human proteome [1,2]. Soluble secreted proteins such as antibodies and protein hormones rely on this pathway. The pathway also delivers transmembrane proteins (TMPs) to the endoplasmic reticulum (ER), its downstream organelles such as the Golgi apparatus, and the plasma membrane.

All secretory proteins are translated by cytosolic ribosomes and must be first targeted to and then transported across (or inserted into) the ER membrane at the early stage of their life, either co- or post-translationally [3,4]. A complex network of cytosolic and ER membrane-resident macromolecules facilitate and assist the ER targeting and translocation.

Both ER targeting and translocation/insertion critically depend on so-called signal sequences (SSs), short hydrophobic peptide stretches in the amino acid sequence of the newly synthesized proteins that are recognized by the secretory machinery as trafficking signals. While SSs may appear exceedingly simple, they possess a remarkably versatile and complex physiology. Besides the choice of trafficking routes, SSs carry information about translocation efficiency, occurrence and timing of cleavage, and post-targeting functions.

There are four main classes of SSs: (i) cleavable signal peptides (SPs), found on secreted proteins such as insulin and type I membrane proteins such as HLA molecules; (ii) type II signal anchor sequences (SASs), found on single- and multi-pass transmembrane proteins (TMPs) such as the membrane-bound form of tumor necrosis factor (TNF); (iii) type III SASs found, e.g., on Sec61β; and (iv) tail anchors (TAs) found on proteins such as Sec61γ (Figure 1a). Signal peptides (SPs) are by far the most populous class of SSs. In humans alone, there are an estimated >3000 different SP-containing proteins, constituting >10% of the whole proteome. SPs are usually localized within the first 30 amino acids of the coding sequence but can in some cases also be found more internally. The defining trait of SPs is the capacity to be cleaved by the aptly named signal peptidase complex (SPC).

The primary sequence of SPs is only loosely defined. In fact, approximately 20% of randomized sequences can promote protein secretion in yeast [6]. SPs are characterized by a tripartite structure (Figure 1b–d): (i) an often positively charged, N-terminal ‘n-region’ that faces the cytosol; (ii) a short hydrophobic core—most commonly between 7 and 15, but not longer than 18–20 amino acids called ‘h-region’; and (iii) a polar luminal C-terminal ‘c-region’ that contains the scissile bond and must be occupied by short, hydrophobic residues at positions −1 and −3 relative to the cleavage site [7,8]. Initially, SPs are inserted into the ER membrane with the N-terminus facing towards the cytosol (N_in_) and the mature sequence facing the organellar lumen (C_out_). In the case of type I TMPs, the removal of the SP leads to an ‘inverted’ topology of the mature sequence in which the N-terminus is facing ‘outside’ (N_out_), while the C-terminus is facing the cytosol (C_in_) (Figure 1a).

In this review, we delineate the lessons learned from the structural characterization of the different secretory machineries and their interactions with SPs, starting at the ribosomal exit tunnel and ending in the ER membrane. For the targeting and translocation of non-SP SSs, we refer to several excellent recent reviews [9,10,11].

## 2. SP Recognition in the Cytosol and ER Targeting

As the first step of their life cycle, secreted proteins and TMPs must be targeted to the ER membrane (Figure 2). The timing of translation, folding, and ER transport is of particular importance: on one hand, nascent chains (NCs) can only cross the ER membrane in an unfolded state, while on the other hand, this prerequisite exposes their hydrophobic segments and makes them prone to aggregation and proteolysis. Therefore, these proteins must be shielded from the hydrophilic cytosol, which is achieved by one of two separate strategies: (i) direct recognition of the nascent protein at the ribosomal exit tunnel by the SRP, leading to co-translational translocation/insertion through the recruitment of the ribosome–nascent chain complex (RNC) to the ER; or (ii) post-translational transport to the ER, which requires the involvement of chaperones to protect the clients from the aqueous environment.

In this section, we dissect the different cellular pathways for SP delivery to the ER surface (Figure 2). The decision of which routes an NC takes largely depends on interactions between the SP and the respective components of a pathway. It should be kept in mind that these pathways are not necessarily strictly separated, and preferences for one pathway or another may reflect the physiological status of the cell [12]. Additional pathways exist for the targeting of other SSs, such as the TRC/GET pathway that mainly caters to TA proteins [13].

### 2.1. Life in the Fast Lane: Co-Translational Recognition and ER Targeting by the SRP

Co-translational targeting of SP-containing proteins is accomplished by the signal recognition particle (SRP) and its cognate ER-associated SRP receptor (SR). The SRP is one of the earliest factors that scan the SPs of NCs when they emerge from the ribosomal exit tunnel. The SRP competes for NC binding with the nascent chain associating complex (NAC) [17]. Thus, whether an SP can interact efficiently with SRP is one of the main determinants of their trafficking route.

The SRP pathway is by far the best-studied ER delivery pathway. It is also considered to be the most common ER delivery system in mammals, although potentially less dominant than earlier assumed [18,19,20,21]. Proteins of the SRP pathway generally have strongly hydrophobic, N-terminal SPs. As a general rule, the less hydrophobic an SP, the less it tends to depend on SRP [22,23,24,25]. However, the dependence of the SRP system on hydrophobicity seems to vary considerably between organisms [21]. In mammals, the majority of SPs are thought to rely on SRP [26], while in yeast, only ~14% of cleavable SPs (which possess shorter hydrophobic segments [5,27]) are SRP-dependent. In contrast to SPs, the overwhelming majority of SASs cannot be efficiently targeted to the ER without SRP, regardless of the organism [21,28,29,30]. In addition to hydrophobicity, the SRP pathway is sensitive to the localization of the SS within the NC, which is mostly relevant for other types of SSs. Because SRPs are strongly sub-stoichiometric compared to ribosomes [31], substrate recognition needs to be fast. Therefore, the optimal SRP recognition window is within the first ~95 amino acids (AA) of the substrate, where the SP is positioned in most pre-proteins [32]. Remarkably, however, internal SSs located up to ~65 AA distal from the pre-protein’s C-terminus can still be recognized by the SRP [21,29]. In contrast, small proteins below 100 AA cannot be efficiently recognized. In both cases, when the SS is too close to the C-terminus or the protein is too short, the protein synthesis is terminated in mere seconds, which is insufficient for SRP targeting [18,33].

Robust targeting by the SRP pathway is estimated to require approximately 5–7 s [34]. When SRP binds to an SP at the ribosomal exit tunnel, its main functions are (i) to halt the translation to prevent cytosolic aggregation, (ii) to recruit RNCs to the ER surface via the SRP receptor (SR), and (iii) to facilitate the handover of the NC to the Sec61 translocon (Figure 2).

The SRP, a composite of one RNA and six protein molecules, binds ribosomes in a ‘scanning’ mode even before SPs emerge from the exit tunnel [29,35,36,37,38]. In some cases, ribosome–mRNA complexes have been found to recruit the SRP before the SP has even been translated [29], or from within the exit tunnel through an allosteric interaction if the N-terminal SS is particularly hydrophobic, which is mostly the case for SASs [39,40].

One of the SRP proteins, SRP54, contains the hydrophobic SS binding groove in its C-terminal M domain. In the scanning mode, the M domain is pre-positioned in direct proximity to the exit tunnel in an auto-inhibited state [41], in which its C-terminal helix ahM6 [14] occupies the SP binding site. As soon as an appropriate SP emerges from the ribosome exit tunnel, it is embraced by the SRP54 M domain in the ‘cargo recognition’ state, providing shelter from the aqueous environment. The mammalian SRP54–SS interface in the ‘cargo recognition’ state has been resolved in a recent structure of SRPs in complex with ribosomes translating a type II SAS with a 16 AA hydrophobic core [14] (Figure 3a–e). The ahM6 helix is displaced by the SS such that it completes a ring of α-helices around the SS helix, which remains accessible at both ends. The bulk of the interaction is mediated through non-directional, hydrophobic protein–protein contacts (mostly through flexible methionine side chains) of the SS core with the SRP54 M domain. The hydrophobic surface of the SRP54 M-domain is about 20–23 Å long, matching hydrophobic sequences of ~16 AA (Figure 3e). It appears plausible that SPs with much shorter h-regions [5,27] produce a partial hydrophobic mismatch and are less efficient in both recognition and the induction of the necessary conformational rearrangements in SRP. Indeed, deletions in the h-region of the (already short) SP of preprolactin have drastic effects in its ability to trigger both SRP binding and ER translocation [42]. Coincidentally, these shorter hydrophobic sequences also reduce the susceptibility to aggregation in the cytosol, rationalizing their reduced dependency on the pathway [21].

The SRP54 M-domain provides the plasticity to accommodate a broad variety of SPs. Positively charged residues in the flanking SP regions fine-tune SRP binding [42]. Indeed, a lower-resolution structure of the rabbit ribosome–SRP complex loaded with a 20 AA type II SAS [35] and crystal structures of homologous SRP/SS complexes [43,44,45,46] display differing orientations of the SS. Whether these models reflect the physiological variability of the SS binding or are affected by modeling inaccuracies or crystal packing artefacts must be addressed in more extensive sampling of SS space.

The formation of the SRP54–SP interface not only increases SRP/RNC affinity [40,47] and slows down translation [48,49,50], but it also enables efficient recruitment of the ER-bound SR [41]. Information of SP engagement is passed within SRP54 by the so-called fingerloop of the M domain to the N-terminal GTPase (NG) domain, which contains the SR interface as well as the GTPase site [14]. The SRP54 NG domain can now dimerize with the NG domain of the SR α-subunit (SRα) at the cytosolic surface of the ER [51], leading to a large-scale conformational change when the GTPase activity of these NG domains is activated [52,53,54]. This large-scale remodeling involves the withdrawal of the SRP54 helices ahM5 and ahM6 from the SP interface, resulting in the partial exposure of the SP to the aqueous environment [14,55]. This conformational change enables cargo handover to the protein conducting channel Sec61, which form the core of ER translocon complex [56]. The interaction kinetics of SRP and SR are regulated by NAC, which prevents the recruitment of SS-less ribosomes to the ER [57].

Faulty SSs, e.g., segments with deletions in their hydrophobic core, trigger a recently discovered quality control pathway, termed regulation of aberrant protein production (RAPP) [58]. As shown by photo-crosslinking, the SRP likely hands over aberrant SPs to Argonaute2 (Ago2) instead of Sec61, triggering Ago2′s specific mRNA degradation. However, there is yet no evidence for a direct interaction of Ago2 with SPs.

### 2.2. With a Little Help from My Friends: Post-Translational Recognition and ER Targeting

Substrates that are targeted to the ER post-translationally require ATP-dependent chaperoning in the cytosol. In yeast, post-translational ER targeting of SP-containing proteins is common [21]. Some ‘classic’ yeast post-translational substrates are targeted co-translationally in mammals, indicating a preference of the co-translational mode in these species [26]. Nevertheless, also in mammals, SRP-independent translocation of non-TA proteins goes much beyond early discovered single cases [59], as recent studies show [18,19,20,33,60,61]. We are still at an early point of our understanding of post-translational import of SP-containing pre-proteins in mammals.

#### 2.2.1. Stairway to Lumen: Hsp70/40-Mediated Chaperoning in the Cytosol and ER Delivery

The Hsp70/40 system is thought to deliver its substrates mainly to Sec61/62/63. Post-translational substrates are usually poor substrates of the SRP, and their structural motives are therefore reciprocal to those of SRP substrates—the less hydrophobic an SS, and the closer it is to the C-terminus, the more likely it is to take a chaperone-mediated post-translational delivery route to the ER [24,62]. Hence, most substrates of the pathway feature SPs, although some type II TMPs and multi-spanning TMPs are also influenced by the depletion of both Sec62 and Sec63. Prominent substrates are weakly hydrophobic, N-terminal SPs with weak statistical trends for long h-regions and low-polarity c-regions [62,63]. The post-translational pathway can process more proteins per time than the SRP-dependent route [12].

Hsp70-mediated ER targeting has so far mostly been studied in yeast. Cytosolic Hsp70 (Hsc70 in mammals, Ssa1 in yeast) has broad substrate selectivity for a degenerate, hydrophobic 4–5 AA motif [64] that occurs in SPs, but also in virtually any other protein (Figure 3f,g). Despite its broad functionality as a chaperone, the protein can carry out substrate-specific ER delivery of secretory pathway proteins. Although Ssa1 can directly interact with Sec72, a part of the fungal Sec62/63 translocon that does not exist in mammals [65], the targeting is largely regulated in a substrate-dependent way by a network of Hsp40 (or J-domain protein) co-chaperones (Figure 2) such as the abundant, ER-tethered Ydj1 (Hdj2 in mammals) or Jjj3 [66,67,68,69,70]. Hsp40s (i) increase the chaperoning activity of Hsp70s [71], (ii) recognize and deliver substrates to their respective Hsp70, and (iii) target their Hsp70 to a specific location in the cell [72]. It appears that these chaperones have some redundancy, as no single one of them is essential [67]. Hsc70-mediated targeting in mammals has been demonstrated for low-hydrophobicity TA proteins, but not yet for SP-containing proteins [73].

In addition to Hsp70/40s, calmodulin mediates targeting in mammals, by recognizing short substrates through their SPs [60]. Proteins of less than 100 AA, often chemokines or hormones, are poor SRP substrates because of their short recognition window at the ribosome [18]. Simply extending their mature sequence has been shown to revert substrates to SRP-dependence [33,74]. The calmodulin-mediated pathway is better characterized for the delivery of weakly hydrophobic TA proteins to the EMC [75].

#### 2.2.2. The Joker: The Snd Pathway

The list of ER targeting pathways was expanded when a screen in yeast revealed three proteins called Snd1-3 that can compensate for the known pathways and act as a backup ER targeting system [15]. The Snd pathway has been implicated in the biogenesis of short secretory proteins [76] as well as both single- and multi-pass transmembrane proteins [16,77]. The system appears capable of recognizing SSs at any position in the protein, from N- to C-terminus [15], allowing it to substitute SRP and other pathways, such as the TRC/GET pathway.

Of the three known yeast Snd proteins, Snd1 was found to be soluble and ribosome-interacting [78], while the other two are ER-resident membrane proteins. Snd2 can be co-immunoprecipitated with Snd1, Snd3, and Sec61, leading to the hypothesis that Snd1 may be involved in target recognition at the ribosome, while the other two subunits could act as a composite receptor and mediate substrate-handover to the Sec61 translocon for translocation, analogous to the SRP/SR system [76]. To date, only the orthologue of Snd2 (called TMEM208 or hSnd2) has been identified in mammals [16,79]. AlphaFold predictions of yeast Snd2/3 show a hydrophilic vestibule on the hydrophobic segments of both proteins (Figure 3h,i), and it is tempting to speculate that these might be key to their function.

## 3. Protein Translocation and Insertion at the ER Membrane

Following ER targeting, secretory pathway proteins must be either translocated through or inserted into the ER membrane. The main insertase/translocase for SPs is the protein-conducting channel Sec61, which is a part of the ER translocon and can associate with different auxiliary protein complexes (Figure 4) [80].

The ER translocon is a dynamic super-complex at the ER membrane that facilitates the translocation, folding, and post-translational modification of many NCs (Figure 4). In addition to its central component Sec61, the ER translocon features a portfolio of sub-complexes with specific auxiliary functions that can be proactively recruited in a sub-stoichiometric manner depending on the type of SS at hand [11,80]. Some, such as the translocon-associated protein complex (TRAP), are associated near-stoichiometrically to the co-translational ER translocon (Figure 4), while others such as the SPC can act in concert but are not or only transiently recruited [81]. New configurations of the translocon are still being discovered and structurally characterized, giving a plethora of new information their interactions with SSs [5,82,83,84,85,86,87,88,89,90]. For example, a range of newly characterized ER-resident insertase complexes containing an Oxa1 family subunit can facilitate the insertion of other types of SSs, particularly type III SASs and TAs, but also multi-pass TMPs (reviewed in [9,10]) in concert with Sec61. The ER-resident Oxa1 family complexes comprise the ER membrane complex (EMC), WRB/CAML, which is the insertase of the TRC/GET pathway, and the TMCO1 translocon. TMCO1 proteins might be involved in the translocation of SP-containing type I membrane proteins by facilitating the insertion of downstream TMDs after initial recognition by Sec61, respectively, while the EMC is fully dispensable for the insertion of SPs and type II TMDs but may have chaperoning activities [75,91]. To date, it appears that Sec61 is the only insertase for SPs.

In the following section, we will discuss how different components of the ER translocon interact and are recruited by features in the SP, starting with its prime component, the protein conducting channel Sec61.

### 3.1. Through the Barricades: Co-Translational Translocation/Insertion

In the co-translational mode, the ER translocon associates with an RNC. The interaction of the RNC with the Sec61 core is conserved from prokaryotes to eukaryotes, while many accessory factors, which interact with Sec61 and often also directly with the ribosome, have evolved. Importantly, such interactions are often competitive, making the ER translocon dynamic.

#### 3.1.1. Tunnel of Love: Protein Translocation by Sec61

The heterotrimer Sec61, consisting of Sec61α, Sec61β, and Sec61γ, is chiefly responsible for (i) co-translationally translocating or inserting nascent chains and for (ii) determining substrate topology [94,95,96]. Its largest subunit, Sec61α, features two pseudo-symmetrical halves (formed by TMD1-5 and 6-10, respectively) that together produce an hourglass-shaped pore across the ER membrane (Figure 5) [92,95,97,98]. The channel must be primed and activated by a partner to enable translocation or membrane insertion. Sec61 is gated in two directions: (i) opening of the two halves in a clam-like motion reveals a lateral gate to the ER membrane; (ii) the same motion also opens the vertical channel that is initially sealed by the ‘plug’ from the lumen, allowing the passage of the nascent chain [99] (Figure 5c,d).

In its co-translational mode, Sec61 can recognize type II SASs in addition to SPs. Upon RNC binding, the gap between the ribosome exit tunnel and the Sec61 translocating channel is reduced to approximately 20 Å [81,92]. The hydrophobic SPs approach the channel head-first. The docking induces conformational changes in Sec61α that, with one rigid body motion, simultaneously open the lateral gate (allowing the SP to squeeze between TMDs 2 and 7 and displace TMD2), and remove the ‘plug’ to vertically open the channel [92,95,97,98].

Subsequently, the NC re-organizes into a hairpin that exposes the N-terminus to the cytosol. The final orientation of the SP inside Sec61 determines the topology that the protein will ultimately assume. The outcome seems to be mainly affected by the balance of positive charges on both ends of the c-region, with positive charges at the n-region encouraging insertion with the N-terminus facing the cytosol and positive charges in the early mature region apparently impairing translocation altogether [97]. For both type I and type II membrane proteins, the N-terminus of the SS faces the cytosol according to the ‘positive-inside‘ rule, with the distinction that the SPs of type I membrane proteins are later removed by the SPC, whereas the signal anchors of type II membrane proteins are not (see below). Nevertheless, positive charges of the n-region are not the only orientation-determining parameter as ~20% of ~900 analyzed SPs do not contain positively charged residues [5,8].

The decisive parameter for Sec61-dependent translocation is the ‘gating’ strength of the SP, which is the ability to autonomously induce the opening of Sec61‘s lateral gate. The gating strength can be measured directly as a pulling force on the nascent chain that drags the SP into the primed channel [100,101]. Weakly gating SPs may trigger the recruitment of assisting co-factors (see below). Interestingly, the gating strength is in many instances evolutionarily conserved, implying that strong or weak gating can fulfill important functions, e.g., in the timing and overall production levels of substrates, or even to maintain a cytosolic presence of the substrate, as is, e.g., the case for calreticulin [102,103]. Despite being measurable, the precise individual structural/physical features that determine the gating strength are poorly understood [104]. The hydrophobicity of the SPs core is suspected to play a key role and other potential factors might be the charge of adjacent residues in the nascent chain, the degree of N-terminal folding, and the lipid composition of the ER membrane [19,105].

When finally opened, Sec61 allows the co-translational translocation of the nascent chain’s hydrophilic regions across the membrane and into the ER lumen, driven by the GTP hydrolysis of the ribosome.

During translocation, the SP of the nascent peptide remains accommodated in the lateral gate of the Sec61 protein-conducting channel [92,98,106] in a slanted orientation. The interaction is mediated by hydrophobic residues and halfway open to the surrounding lipid environment (Figure 5) [92]. Early crosslinking experiments showed that changing the amino acid sequence in the h-region can change the positioning of the SP in the gate [25], implying that the interaction is subject to considerable conformational liberty.

Interestingly, electron tomography data of pancreatic vesicles suggest that SPs seem to linger stably at the lateral gate while the ribosome is bound to the ER translocon because Sec61 is detected solely in an open state with SPs bound [92,106]. This observation suggests that SPs are not able to diffuse freely into the membrane, presumably due to their short h-regions that cause a hydrophobic mismatch with the ER bilayer. In the case of cleavable SPs, the peptides eventually need to be transferred from Sec61 to the SPC, although it is currently unclear how this handover may take place.

#### 3.1.2. Smooth Operator: TRAP Is a Sec61 Assistant

SRP delivers SSs to Sec61 irrespective of their ‘gating‘ efficiency [100,107]. When encountering substrates with ineffective or slow channel gating abilities, Sec61 requires assisting factors that help it to exert second pulling event that occurs during translocation. The most stoichiometrically present of several auxiliary complexes is TRAP, which specializes in co-translational enhancement of Sec61 translocation/insertion.

Some of the features of SPs that mediate important functions—such as non-standard lengths, charged residues, kinks, or reduced hydrophobicity—are also hallmarks of reduced ‘gating‘ efficiency and therefore particularly challenging for Sec61-mediated translocation/insertion. It appears that no single parameter determines TRAP dependency—rather, it is the combination of several parameters that ultimately weaken the interaction with the primed Sec61 translocating channel [104]. Although there are subtle differences, unifying characteristics of TRAP-dependent substrates are (i) a low hydrophobicity and/or (ii) positively charged clusters in the mature portion directly following the SP that impair translocation [62,63,108], and (iii) an enriched glycine and proline content [63,109]. Additionally, TRAP may be able to assist in the translocation of charged unstructured protein domains [108].

TRAP comprises the four subunits TRAPα-δ (or SSR1-4) [104]. The helix-breaking properties of TRAP-dependent substrates may impair their ability to insert into the lateral gate of Sec61α [92], potentially resulting in SPs that ‘lounge’ in the ~20 Å gap between ribosome and Sec61 instead of being pulled in by primed Sec61. TRAP is associated near-stoichiometrically with the translocon and interacts with both the ribosome and Sec61 [81,90,106,110].

Mechanistic insights into TRAP function remain scarce. A recent study delineated the ‘translational timeline’ of TRAP-assisted translocation. Based on crosslinking experiments it postulates a weak, potentially sensory interaction of the ~13-amino-acid-long cytosolic portion of TRAPβ with a weakly gating model SS before the SS is threaded into the Sec61 channel [109]. Tomographic studies [63,81] indicate that the luminal domains of TRAPα/β might interact with loop 5 of the Sec61 hinge region to facilitate channel opening. Additionally, it has been suggested that that luminal domains of TRAP may stabilize the topology of thus far ‘uncommitted’ nascent TMDs by a direct interaction [111]. However, there are currently no structural data of sufficient resolution to corroborate these postulated interactions.

### 3.2. Insane in the Membrane: Post-Translational Translocation/Insertion

Much like the co-translational pathways, post-translational translocation/insertion of proteins with N-terminal SPs and SSs is carried out through Sec61 (aided by auxiliary factors) while Oxa1 family complexes such as EMC and Get1/2 complexes likely do not interact with SPs.

#### With Arms Wide Open: Post-Translational Insertion and Translocation Assisted by Sec62/63

Some secretory proteins are targeted to Sec61 post-translationally [18,24,67,74,112,113,114]. Together, the auxiliary proteins Sec62 and Sec63 can facilitate post-translational Sec61-dependent translocation/insertion, allowing Sec61 to translocate substrates by prying open the lateral gate [115,116,117,118]. First, Sec63 partially opens the gate through interactions on both sides of the membrane, before Sec62 can fully displace the plug and shield the channel from incoming lipid molecules [84,93,119,120].

To date, 56 Sec62/63 dependent proteins have been identified in human cell lines [62,63]. The SP features that confer Sec62/63 dependency are very similar to those for its Hsp70/40 delivery system, mainly featuring reduced hydrophobicity and weak Sec61 gating capacities, which make the pre-opening of the channel by Sec62/63 necessary [120]. Additionally, some Sec62/63-dependent substrates possess a positively charged cluster in the mature region that likely impairs the translocation across the Sec61 channel [62]. Notably, there is some overlap between Sec62/63- and TRAP-dependent SSs, as both comprise low-hydrophobicity cores. Indeed, there are specific examples that have been shown to depend on both complexes [62,63], although TRAP is thought to operate co-translationally whereas Sec62/63 is generally described to act post-translationally. It has been reported that some substrates can follow both the co- and the post-translational route, such as pre-proinsulin [121].

The Hsp70/Hsp40 system also possesses a luminal portion, which is essential for post-translational ER translocation by providing directionality and might also play a role during co-translational import. In absence of a ribosome, protein translocation is powered from within the lumen by the Hsp70 ‘ratchet’ BiP (HSPA5, or Kar2 in yeast), which is recruited by the Sec63 (or ERj2) Hsp40 J-domain and can interact with a luminal loop of Sec61α [76,122,123,124,125]. Another example is the ER membrane-resident mammalian J-protein ERj1 (Htj1 in humans) can bind ribosomes near the ribosomal exit tunnel using its cytosolic portion, promote translational stalling, and potentially recruit ribosomes to the ER during co-translational targeting. Its J-domain is located in the ER lumen, where it aids in protein translocation by stimulating BiP, similarly to Sec63 [126]. Other ERj proteins modulate BiP functions for protein quality control or stress conditions such as the unfolded protein response or ER-associated protein degradation [127].

Some studies of specific substrates found crosslinks of mammalian Sec62/63 with the co-translational ER translocon, implying that Sec62/63 might also be involved in co-translational protein translocation [128,129,130]. However, structural studies of the early co-translational ER translocon suggest that there would be steric clashes between Sec61-bound Sec62/63 and both the ribosome and TRAP [81,95]. Thus, either the conformation of Sec61/62/63 must differ in the co-translational mode or the ribosome associates differently [84,120,131]. Biochemical studies corroborate the competitive binding of co- and post-translational ER translocon components by showing that both the SR and the ribosome individually induce the dissociation of Sec62 from Sec61 [131].

Together, these data suggest that Sec62/63 might only be involved in the later stages of co-translational translocation, if at all [129]. As an exemplary hypothesis for such an operating mode, TRAP might initially ensure the engagement of the SP by Sec61, before Sec62/63 eventually carry out in-channel inversion and gating in a temporally distinct step after translocon remodeling [108]. However, there is currently no mechanistic basis for these kinds of proposed mechanisms, and it is not known if and to what extent Sec62/63 can compensate for a loss of TRAP or vice versa.

## 4. SP Removal and Post-Targeting Functions

### 4.1. Time to Say Goodbye: SP Removal by the Signal Peptidase Complex

Regardless of how the specific characteristics of SSs influence the routes taken, the ultimate choice between SPs and SASs is—per definition—made after ER targeting and translocation. Completed translocation of about 80 amino acids through Sec61 marks the earliest possible time point for co-translational SP removal [132]. The SPC, a hetero-tetrameric serine protease, which exists in two distinct paralogs in higher eukaryotes [5], is responsible for this process.

By definition, the SPC only cleaves SPs, although TAs can also be engineered sufficiently short to be cleaved in the ER lumen [133]. SPC substrates require an h-region below 18–20 amino acids and a c-region with short, hydrophobic residues at the positions −1 and −3 relative to the cleavage site [5,133,134].

Although it is assumed that SP removal typically occurs co-translationally, post-translational SP cleavage is well within the SPCs repertoire [135]. SPs of co-translationally translocated NCs need to transfer from Sec61 to the SPC. However, the SPC is not resolved as part of native ribosome–translocon complexes by cryo-electron tomography [106,136], indicating that the SPC likely associates with the ER translocon transiently or in a structurally flexible manner [81].

The molecular workings of eukaryotic SP removal, and the distinction between SPs and SASs by the SPC, have eluded mechanistic explanation for decades. Recent structural insights reveal that the enzyme complex employs two structural motifs to make this distinction [5]: (i) a specialized transmembrane window (TM window) that locally thins the ER bilayer directly above (ii) a conserved shallow hydrophobic binding pocket that contains the cleavage site, located in the lumen about 17 Å from the luminal membrane interface (Figure 6). Using the TM window as a ‘molecular ruler’, the enzyme measures the length of its substrates’ hydrophobic segments and excludes SSs with TM segments of more than 18–20 amino acids [133,134]. Substrates that have short enough TM segments (h-regions) to be admitted into the binding pocket then adopt a β-strand conformation in the stretch directly adjacent to the membrane (c-region) and are scanned for small, hydrophobic residues at the evolutionarily ancient −1 and −3 positions relative to the cleavage site by the SPC’s catalytic subunit SEC11A/C [137]. Additionally, SPs with prolines at position +1 cannot be cleaved efficiently [138], likely because cis-peptide bonds cannot be effectively polarized in the protease’s oxyanion hole. Based on this bifold recognition principle, the SPC effectively ‘defines’ the distinction between SPs and type II SASs by the length, but not the primary sequence of their hydrophobic segments and by the presence of a suitable set of −1/−3 residues in direct proximity to the ER membrane–lumen interface. Type III SASs cannot be cleaved by the SPC because of their reversed topology that prevent the recognition of the scissile bond.

The timing of SPC-mediated SP removal can impact a protein’s maturation process. Especially viral pre-proteins of flavi- and retroviruses, which heavily depend on the SPC for their maturation, utilize cleavage delay to halt the trafficking of their pre-proteins, which is crucial for maintaining the correct processing order and joint incorporation of the fragments during virion assembly [135]. Delays can be caused by a highly apolar or helical c-region, folded n-regions, or directly neighboring TM segments that may sterically obstruct access to the cleavage site [139,140,141]. It will be interesting to further assess the details of the SPC-SP interactions by structural biology, and to define the interactions of the SPC with the ER translocon during co-translational SP cleavage.

### 4.2. The Show Must Go On: Post-Targeting Functions of SPs

While SASs and TAs remain associated with their proteins, SPs can have a range of interesting and versatile functions after their cleavage that are unrelated to their original protein. While many are recycled by signal peptide peptidases, specific SPs can, for example, modulate NK-cell mediated immune responses via HLA-E, become a part of mature virions, or associate with cytosolic factors such as calmodulin as feedback loops. For further reading, we refer to [12,142].

## 5. Summary and Outlook

### 5.1. The Long and Winding Road: Principles of SP Recognition

Amongst the many interactions laid out here, the three key moments for SPs are (i) the efficiency of the interaction with SRP54, which occurs very early and effectively seems to decide whether a protein takes the co- or the post-translational route; (ii) the gating of Sec61, which determines whether auxiliary factors need to be recruited; and (iii) their removal by the SPC, which is usually necessary for the functionality and further trafficking of the protein. The loose sequence requirements for the SP targeting and translocation machinery lead to an elastic and partially overlapping system, in which subtle changes can alter pathway usage. With recent structural data, the molecular rules and responsible mechanisms that determine the SP’s fate are slowly emerging.

There are a few governing principles followed by all or many of the proteins that interact with SPs. Their main interaction point is obviously the h-region. Two major types of strategy emerge for this interaction: (i) cytosolic factors such as the SRP and Hsps bury the h-region in a hydrophobic groove that can be closed by a lid. The interface is often lined by dynamic residues such as methionine and does not induce a specific shape requirement for the substrate. Therefore, it is rather the absolute hydrophobicity and the length of the h-region that determine how efficient the binding is. (ii) Intramembrane interfaces, found, e.g., on that Sec61 lateral gate and the SPC, utilize the lipid surrounding and often leave a substantial portion of the h-region surface open to the membrane, which greatly reduces the requirement for shape complementarity.

The resulting low-sequence constraint on SPs provides an optimal opportunity for an evolutionary fine-tuning of beneficial interactions. As such, the ER delivery and translocation route (and with it, e.g., the speed and absolute capacity of protein synthesis [12]), the efficiency and speed of translocation/insertion (and with it, e.g., the maintenance of a cytosolic pool [102,103]), the timing of SP cleavage (influencing, e.g., the time to add posttranslational modifications [135]), and in some cases even the post-cleavage functions can all be used to tailor to the biosynthetic needs of a specific protein. In addition, SPs can be used to generate alternatively localized versions of a protein from a single mRNA. So how are these changes accomplished, precisely? Unfortunately, too many unknowns still exist to answer this question decisively.

Lastly, the recognition principles employed by the SPC are markedly different from those of other complexes involved in targeting and translocation. As a result, the choice of co- or post-translational translocation/insertion does not affect or only moderately affects SPC cleavage. This difference in recognition for the strictly necessary step of SP removal greatly increases the evolutionary freedom for the largely optional targeting and translocation routes.

### 5.2. Imagine: Outlook

It is entirely possible that more ER targeting and translocation pathways await discovery. For example, it has only recently emerged that thinning of the particularly elastic ER membrane is a common principle among ER-resident Oxa1 family protein complexes, which use it to catalyze the insertion specific types of transmembrane helices, and by the SPC, which use it as a ‘molecular ruler’ to measure the length of the h-region [5,10,85]. These findings were rather unanticipated, but have substantially broadened our understanding of the membrane insertion of less well-studied SSs such as type III SASs, TAs, and the downstream TMDs of multi-pass membrane proteins [82,85,143].

There are, however, more gaps in our understanding [144], particularly when it comes to the handover of substrates between complexes, which is one of the most difficult problems to solve with structural biology, and to the relative usage frequency of the individual pathways, which may well differ between organisms and physiological states. As such, relatively few post-translational substrates are known to date. Additionally, the overlap between the pathways greatly complicates the experimental design, rendering a coherent understanding difficult [15].

Compared to yeast, the ER-resident protein translocation machinery has evolved substantially in mammals. It is striking that many genes have duplicated, giving rise to several paralogs at various stages of protein biogenesis in the ER [145]. For example, the mammalian ER possesses two oligosaccharyltransferase paralogs that specialize in co- or post-translational processing, respectively [146], and the SPC exists in two paralogs with overlapping function [5]. Another example would be Sec61α, which exists in two paralogs, the latter of which has been poorly characterized to date. More studies are needed that focus on the quantification of the pathway usage, particularly in mammals, the evolutionary advantages of several paralogs, whether the different paralogs could fulfill specialized roles with respect to SP interactions.

## Figures and Tables

**Figure 1 ijms-22-11871-f001:**
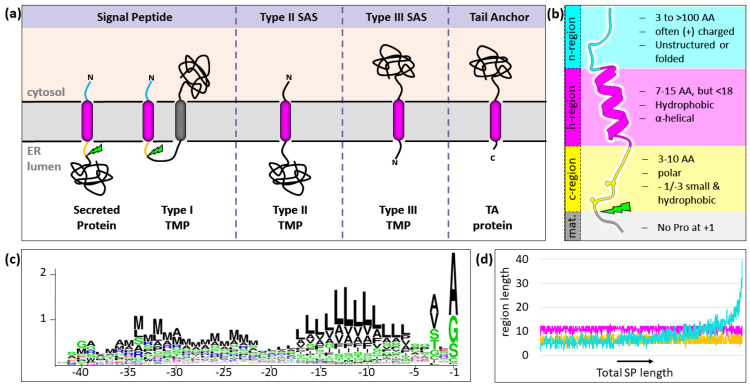
Types of signal sequences. (**a**) Depiction of the four types of SSs with their membrane topology indicated. Hydrophobic segments are depicted in magenta. (**b**) Signal peptides have a tripartite structure, consisting of an n- (cyan), h- (magenta), and c-region (yellow) and are cleaved in the ER lumen by the SPC (green flash). (**c**) Frequency of residue types relative to the cleavage site [5]. (**d**) Relative length of the respective regions (colored as in (**b**)) as a function of total SP length. The bulk of the length variation stems from the n-region. Panels c–d were adapted from [5].

**Figure 2 ijms-22-11871-f002:**
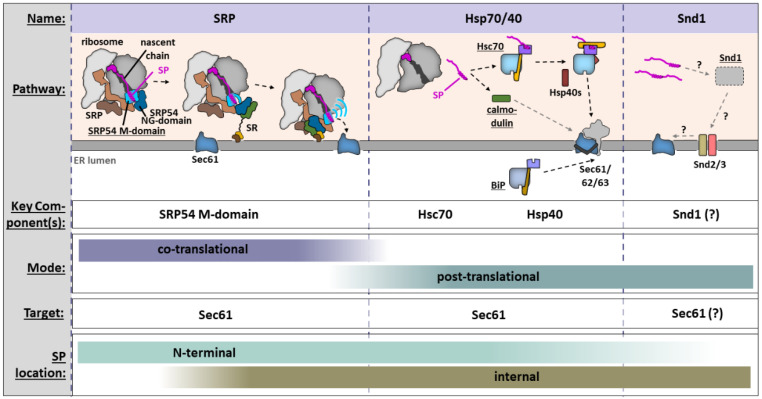
ER delivery pathways for SP-containing proteins. The upper panel shows a schematic of each delivery pathway. The central component of each pathway is underscored. **Left**: SRP (blue/brown subunits) recognizes SPs emerging from the ribosomal exit tunnel and shields them through the SRP54 M-domain. SP binding triggers the heterodimerization of the SRP54 NG domain (blue) with that of SRα (green), guiding the RNC to the ER. A large conformational rearrangement partially exposes the SP for handover to Sec61 [14]. **Middle**: Cytosolic calmodulin or Hsp40-assisted Hsc70 recognize SP-containing proteins and guide them to the ER. **Right**: The recently discovered Snd pathway likely consists of a cytosolic component, Snd1, which might act as a chaperone, and two ER membrane-resident components, Snd2/3, which might facilitate handover to Sec61 in some unknown way [15,16]. Lower panels show specifics of each pathway.

**Figure 3 ijms-22-11871-f003:**
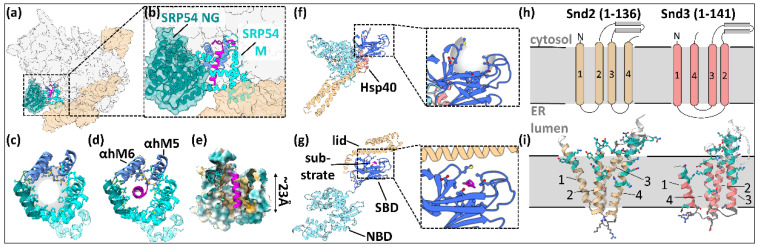
Details of cytosolic SP recognition and ER targeting. (**a**–**e**) The SRP (orange) and its central component SRP54 (teal/cyan) recognize SPs at the ribosomal exit tunnel (**a**,**b**) and bury the hydrophobic core in cavity formed by the SRP54 M-domain (**c**,**d**). αhM5/6 are colored blue. (**e**) Dimensions of the hydrophobic SRP54–SS interface. Figures based on PDB 7OBR [14]. (**f**,**g**) Cargo recognition by Hsp70 and Hsp40 binding, exemplified by bacterial DnaK complexed with the Hsp40 DnaJ in an open conformation [PDB 5NRO] and substrate-engaged high-affinity conformation [PDB 2KHO]. ATP hydrolysis, stimulated by substrate and Hsp40 binding, leads to a high affinity state with a closed lid domain (yellow). NBD: nucleotide binding domain; SBD: substrate binding domain. (**h**,**i**) AlphaFold 2 predictions of the ER-resident membrane proteins Snd2/3 from *S.*
*cerevisiae* (entries Q99382 and P38264). TM helices are numbered Membrane topologies are based on UniProt. Both predicted structured possess hydrophilic vestibules that reach into the presumed membrane environment. Helices are numbered.

**Figure 4 ijms-22-11871-f004:**
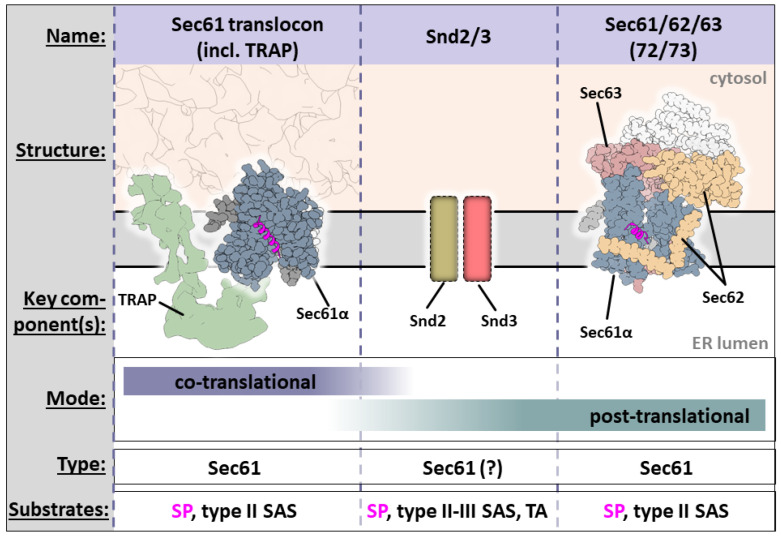
ER translocation and insertion machineries for SP-containing proteins. Left upper panel: The Sec61 translocon comprises, among others, the ribosome (shown in the cytosolic background), Sec61 (Sec61α blue, Sec61β dark gray, Sec61γ light gray), and the yet structurally uncharacterized TRAP (green EM map). The structure is a composite of EMDB 4315 [90] and PDB 3JC2 [92]. Middle upper panel: The yet structurally uncharacterized Snd2/3 membrane complex [15]. Right upper panel: *S. cerevisiae* Sec61 opened by Sec62 (light teal), Sec63 (dark teal), and the yeast-specific Sec71/72 (white). The structure is a composite of PDBs 7AFT and 6ZZZ [93].

**Figure 5 ijms-22-11871-f005:**
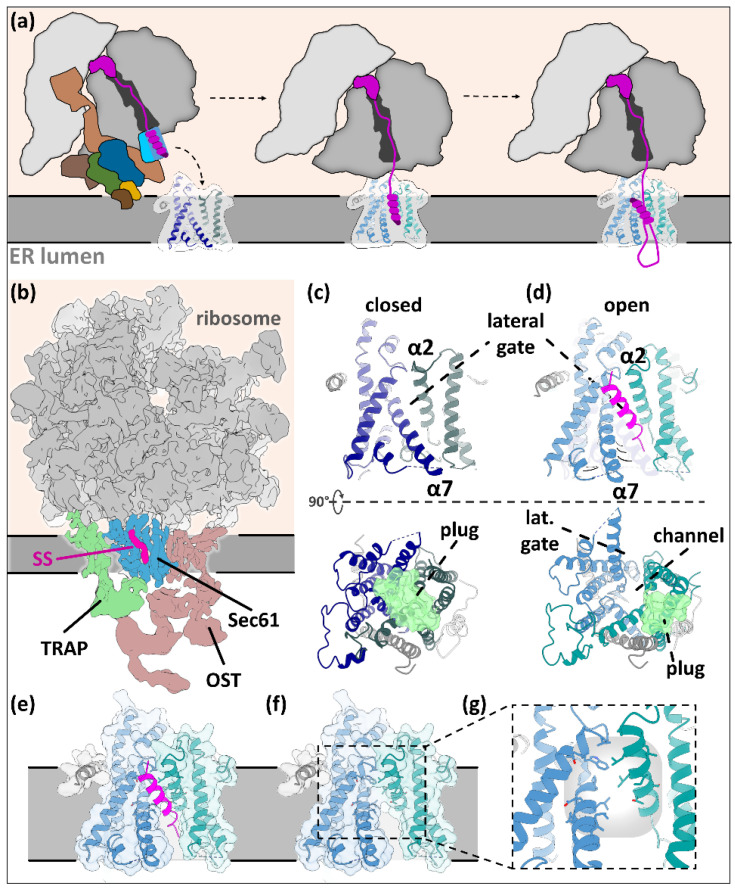
Co-translational ER translocation by Sec61 through the ER translocon. (**a**) Graphic description of co-translational ER insertion/translocation by the SRP-Sec61. The SP is handed over from SRP54-M and inserts head-on into primed Sec61, removing the plug, before rearranging in a hairpin with a N_in_-C_out_ conformation. The SP is then accommodated at the lateral gate of Sec61. (**b**) Tomographic map of the translating ER translocon (EMDB-4315) [90]. TRAP is accommodated at the side of Sec61, its cytosolic portion interacting with the ribosome, its luminal portion close to the opening of the translocation channel. The oligosaccharyltransferase (OST) does not interact with SPs. (**c**,**d**) Open and closed conformations of Sec61, based on PDBs 3J7Q and 3JC2 [92,95]. Upon channel opening, the plug (green, seen from within the lumen facing the membrane) is removed and the vertical channel is opened. The lateral gate can now harbor the SS. (**e**–**g**) SP binding site at the lateral gate. The binding site is partially open to the ER lipid environment, partially formed by hydrophobic residues of helices α2 and α7.

**Figure 6 ijms-22-11871-f006:**
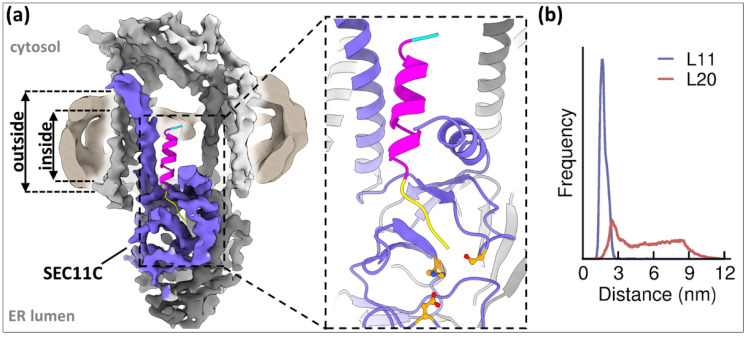
SP removal by the signal peptidase complex. (**a**) Atomic model of an SP fitted into the binding pocket of SPC-C (shown as EM map, catalytic subunit SEC11C purple, SEC22/23 dark gray, SPC25 gray, SPC12 light gray; EMDB-13172). The micelle (show in the background in pale yellow) is thinned inside the TM window and measures the length of the h-region. The blowup shows the binding groove for with the SP c-region and the catalytic residues of SEC11C (orange). Residues −1 and −3 point towards the bottom of the pocket and therefore need to be short and hydrophobic. (**b**) Coarse-grained molecular dynamics simulations show the mean distance of an SP with 11 residues (L11) and 20 residues (L20) from the binding pocket. Long h-regions are excluded by the thinned membrane in the TM window. Figure based on [5].

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
