# Peer review of "Take Me Home, Protein Roads: Structural Insights into Signal Peptide Interactions during ER Translocation"

_ijms, 2021, doi:10.3390/ijms222111871_

Round 1
Reviewer 1 Report
This review is a wonderful overview about the biology of signal peptides. It describes the components that interfere with signal peptides and their mechanisms of action. It is well suited for beginners as well as for advanced readers. The illustrations are excellent and better than in any current text book. The text is very clear, easy to follow and the headlines are really funny. The manuscript is written by leaders in the field. It was a pleasure to read!
I do not see any issue that needs to be fixed. I only found two very minor little details that the authors should correct before this study can be published, basically as it stands:
Page 6, line 210: Ssa1 in yeast instead of SSa1 in yeast
Figure 4. Please indicate which side of the membrane faces the cytosol, which one the ER lumen
Author Response
We thank reviewer #1 for her/his supportive and constructive comments. The manuscript has been revised in the light of those comments. In the following we list all specific comments and address each of them. We are also happy that both reviewers seem to like the song titles!
Answers to the specific comments
Page 6, line 210: Ssa1 in yeast instead of SSa1 in yeast
Done.
Figure 4. Please indicate which side of the membrane faces the cytosol, which one the ER lumen
Done. We also added this info to Figure 6.
Additional changes
In addition to the reviewer comments, we fixed some minor typos, punctuation errors, and imprecise phrasings - all of which are annotated in the revised manuscript.
Reviewer 2 Report
This is a nice summary of the importance of signaling peptides for protein homeostasis regulated by the ER, additionally superbly described for their cytosolic and ER-involved partner proteins.
Figure 1, Skip d) not much information for the reader
Extend size c) a bigger insert makes it more readable and understandable
Figure 2, Post-translational: remove ERj1 not much is known about the function in post-translational transport, a homologue is not known in yeast
Add EMC into the picture
Line 123, Mention NAC in the chapter as ribosome associated factor and putative targeting factor
Line 129, one RNA and six protein molecules
Line 142, Alpha symbol not clear ?-helices
Figure 6, Reduce image b) a bit
Line 566, Sentence: More studies are needed ... try to combine with last sentence (will sound more smooth and we read not to often : need to be done)
The older generation will be happy to read all this song titles again :-)
That´s all
Nice work, concrats!
Author Response
We thank reviewer #2 for her/his supportive and constructive comments. The manuscript has been revised in the light of those comments. In the following we list all specific comments and address each of them. We are also happy that the reviewer liked the song titles!
Figure 1, Skip d) not much information for the reader
We believe that panel d) does convey some useful and often overlooked information: That most of the variability of SPs originates from the n-region, which is the cytosolic portion of SPs, while h-regions are usually of rather constant length. The data is based on a new analysis of >900 experimentally verified SPs. This has e.g. implications for SP interaction with the SPC. Long n-regions are especially prevalent in viral SPs, and we know from experience that it can be difficult to recognize them. SPs with very long n-regions often fail to predict using e.g. SignalP, because long n-regions are currently penalized by the algorithm. We hope this panel may clear things up for the readers. We therefore opted to keep the panel, but reduced it in size in favor of panel c).
Figure 1, Extend size c) a bigger insert makes it more readable and understandable
Done.
Figure 2, Post-translational: remove ERj1 not much is known about the function in posttranslational transport, a homologue is not known in yeast
Done.
Figure2, Add EMC into the picture
We are not sure about the rationale for adding the EMC here, since it has to our knowledge not been shown to insert SPs (although it would be plausible given the EMC's ability to insert TA-proteins). Indeed, the EMC has been shown to be fully dispensable for SP and type I TMD insertion (Chitwood et al., Cell 2018; Ref 92). For this review, we opted to not discuss complexes in detail that do not interact with SPs, mainly the EMC and the GET pathway. We added a short half-sentence to the main text (line 280 ff) that reiterates this notion.
Line 123, Mention NAC in the chapter as ribosome associated factor and putative targeting factor
We thank reviewer #2 for this suggestion and agree to the comment. In addition to the already mentioned function of NAC as a competing factor for SRP on the ribosome (line 110, Ref. 17), we now mention NAC-mediated regulation of the SRP-SR kinetics in line 180, based on Hsieh et al., Nat Comm 2020 (Ref 57).
Line 129, one RNA and six protein molecules
Done.
Line 142, Alpha symbol not clear ?-helices
Done.
Figure 6, Reduce image b) a bit
Done.
Line 566, Sentence: More studies are needed ... try to combine with last sentence (will sound more smooth and we read not to often : need to be done)
Done.
Additional changes
In addition to the reviewer comments, we fixed some minor typos, punctuation errors, and imprecise phrasings - all of which are annotated in the revised manuscript.